# Effect of Biomass Ash on the Properties and Microstructure of Magnesium Phosphate Cement-Based Materials

**Shuguang Zhou [1], Ye Shi [2,\*], Pengtao Wu [2,\*], Haiyu Zhang [1], Yuetong Hui [1] and Wei Jin [1]**

[1] Department of Logistics Supporting, Logistics University of Chinese People's Armed Police Force, Tianjin 300162, China
[2] Tianjin Key Laboratory of Civil Structure Protection and Reinforcement, Tianjin Chengjian University, Tianjin 300384, China
\* Correspondence: 15574300900@163.com (Y.S.); wupengtao0228@tcu.edu.cn (P.W.)

**Abstract:** The disposal of biomass ash (BA) will be of great importance for environmental protection and sustainability, and the aim of this study is to analyze the feasibility of the resourceful use of biomass ash in civil engineering materials. The effects of the content and type of biomass ash on the flowability, setting time, compressive strength, flexural strength, bonding strength, and drying shrinkage of magnesium phosphate cement (MPC) mortar were investigated. In addition, the effects of BA on the hydration and microstructure of MPC were investigated by X-ray diffraction (XRD), thermogravimetry, mercury intrusion porosimetry (MIP), and scanning electron microscope (SEM). The results showed that BA significantly affects the flowability and setting time of MPC mortar. The compressive and flexural strength of MPC mortars decreases with increasing amounts of BA. The drying shrinkage of MPC mortar specimens increases exponentially with the increase of BA content. The incorporation of BA will reduce the bonding strength of the MPC mortar, which is associated with increased drying shrinkage. The incorporation of BA into MPC results in low hydration product generation and poor pore structure. The incorporation of BA into MPC has a significant effect on the microstructure morphology and the hollow columnar-like hydration product may be formed by the reaction of BA with MgO in the paste.

**Keywords:** biomass ash; magnesium phosphate cement; mechanical property; microstructural; hydration

## 1. Introduction

As the world's fourth largest energy source, biomass energy has an extremely important impact on human development and the climate emergency [1–3]. However, the process of burning agricultural and forestry biomass to produce energy will also leave behind a large number of by-products, known as biomass ash. For example, China will generate more than 800 million tons of BA each year from biomass energy. Improper disposal of biomass ashes, such as simple stockpiling, can damage the soil structure, pollute surface water and pollute the air, which can seriously damage the ecological environment [4]. Therefore, how to effectively dispose of biomass ash to prevent it from damaging the ecological environment has become an important research element in the sustainable development of biomass energy. There are two main ways for the disposal of biomass ash: firstly, using it as a fertilizer for agricultural land and forests; secondly, using it as a partial filler material for engineering materials. However, biomass ash used as a fertilizer poses a catastrophic risk to human health. Biomass ash contains a large amount of active $SiO_2$, which has a good pozzolanic reactivity [5]. Therefore, the use of biomass ash as a supplementary cementitious material in concrete will be the main method of utilization, which will serve as a good solid waste resource utilization and achieve sustainable development.

Magnesium phosphate cement is a new type of cementitious material consisting of magnesium oxide, phosphate, and retarder as the main raw materials [6,7]. The hydration reaction of MPC is significantly different from that of Portland cement, which results in

high early-age strength, good bonding properties, low drying shrinkage, and corrosion resistance [8–10]. Due to such performance characteristics, MPC has been widely used in the field of structure rehabilitation [11,12] and nuclear waste stabilization [13]. However, magnesite, the main material for MPC, is a non-renewable ore resource and is complex to process, resulting in high economic and environmental costs [14]. At the same time, the high exothermic heat of hydration of MPC tends to cause problems of temperature cracking [15]. These problems will therefore limit the use of magnesium phosphate cement in engineering. To overcome these problems, one effective method is to utilize industrial or agricultural solid wastes as an admixture for MPC [16,17].

FA and SF, which can react with MPC to produce magnesium silicate gel due to the high $SiO_2$ content, are the commonly used admixture in MPC [18,19]. Those materials have a high utilization rate as they are already widely used in concrete, which results in their relatively high price and the difficulty of obtaining quality raw materials. Exploring new solid waste materials as admixtures in MPC has high social and economic value. Considering that BA has a large stock, low utilization rate, and high $SiO_2$ content [5,20], it has the potential to be used as an admixture for MPC. The application of BA in MPC could be treated as a simple way to reduce the economic and environmental costs of MPC. The application of biomass ash as an admixture in concrete to replace Portland cement has already achieved some success [21–25]. Up to now, there is only a little research on the application of a BA in civil engineering. The effect of BA on the hydration and performance of MPC still needs further research, which will guide the design of BA-MPC materials.

Based on the premises mentioned above, this study systematically investigated the effect of the content and type of BA on the flowability, setting time, compressive strength, flexural strength, bonding strength, and drying shrinkage of MPC mortar. Meanwhile, the effects of BA on the hydration and microstructure of MPC were investigated by XRD, TG-DSC, MIP, and SEM.

## 2. Experimental Details

### 2.1. Raw Materials and Sample Preparation

MPC was composed of dead-burned MgO, $KH_2PO_4$, $NH_4H_2PO_4$, and $H_3BO_3$. The density and specific surface area of dead-burned MgO were 3.42 $g/m^3$ and 251 $m^2/kg$, respectively. The $KH_2PO_4$, $NH_4H_2PO_4$, and $H_3BO_3$ were industrial-grade products with a purity of >95.0%. The BA was collected from the Tianjin Biomass Power Plant and then sieved out of powder form by a screen. Two kinds of BA were used in this study (as shown in Figure 1). One grey BA was named FBA. The other is in black powder form, known as BBA. The density and specific surface area of FBA were 2.484 $g/m^3$ and 269 $m^2/kg$, while 2.462 $g/m^3$ and 253 $m^2/kg$ for BBA. The chemical composition of dead-burned MgO and BAs by XRF is shown in Table 1. Figure 2 provides the particle-size distributions of BAs by laser diffraction method. Anhydrous ethanol was used as the dispersant for particle-size distribution text. The natural sand with fineness modulus 2.0 was used as fine aggregate.

**Table 1.** Chemical composition of dead-burned MgO and BAs as determined by XRF.

| Compounds | Dead-Burned MgO | FBA | BBA |
|---|---|---|---|
| MgO (%) | 96.76 | 5.24 | 5.18 |
| $SiO_2$ | 0.72 | 52.15 | 49.38 |
| $Al_2O_3$ | 0.26 | 11.72 | 9.5 |
| CaO | 1.34 | 9.73 | 9.63 |
| $K_2O$ | - | 4.73 | 4.61 |
| $Fe_2O_3$ | 0.76 | 4.61 | 4.59 |
| $Na_2O$ | - | 2.44 | 2.48 |
| $P_2O_5$ | - | 1.58 | 1.63 |
| $SO_3$ | - | 1.55 | 1.4 |
| $TiO_2$ | - | 0.94 | 0.93 |
| LOI | - | 1.3 | 1.9 |

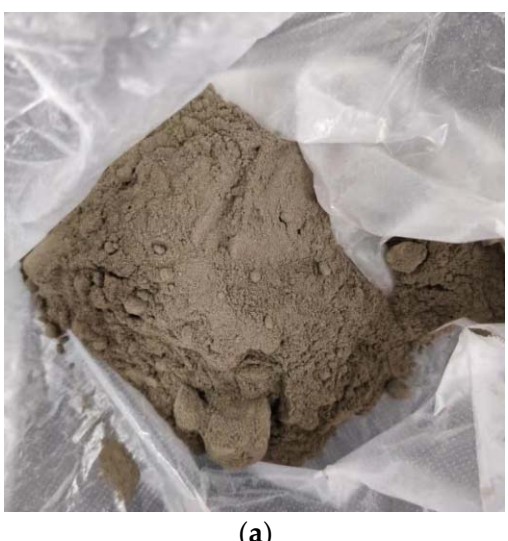

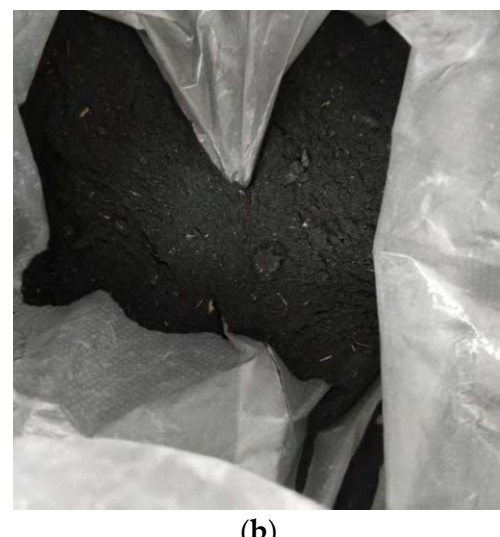

**Figure 1.** Images of FBA (**a**) and BBA (**b**).

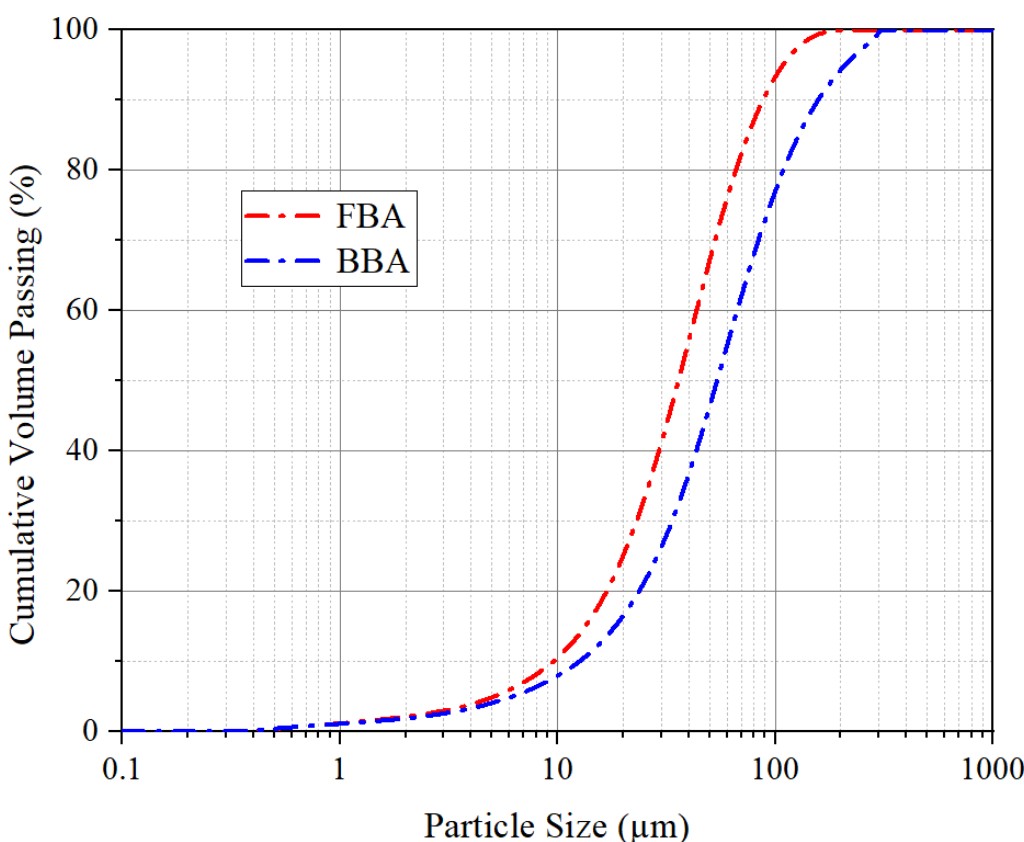

**Figure 2.** Particle-size distributions of Bas by laser diffraction method.

The mix ratio for the control group was determined according to the previous experimental results [4]. A series of BA-MPC mortars were designed, as shown in Table 2. The preparation process of the mortar specimen is shown as follows: first, all powder materials were mixed in the dry state for 1 min; second, water was added for mixing for 3 min; finally, the fresh mixture was cast into a mold and sealed with plastic sheets. After demolding, the specimen was cured in the laboratory at 20 ± 2 °C and 55% RH. The pictures of specimens for different mix proportions were shown in Figure 3.

**Table 2.** Mix design of BA-MPC mortars composites (kg/m$^3$).

| NO. | MgO | NH$_4$H$_2$PO$_4$ | KH$_2$PO$_4$ | H$_3$BO$_3$ | BA | Sand | Water |
|---|---|---|---|---|---|---|---|
| 100 MPC | 780 | 156 | 117 | 90 | 0 | 1143 | 183 |
| 5 BA-95 MPC | 741 | 148 | 111 | 86 | 57 | 1143 | 183 |
| 10 BA-90 MPC | 702 | 140 | 105 | 81 | 114 | 1143 | 183 |
| 15 BA-85 MPC | 663 | 133 | 99 | 77 | 171 | 1143 | 183 |
| 20 BA-80 MPC | 624 | 125 | 94 | 72 | 228 | 1143 | 183 |
| 25 BA-75 MPC | 585 | 117 | 88 | 68 | 285 | 1143 | 183 |

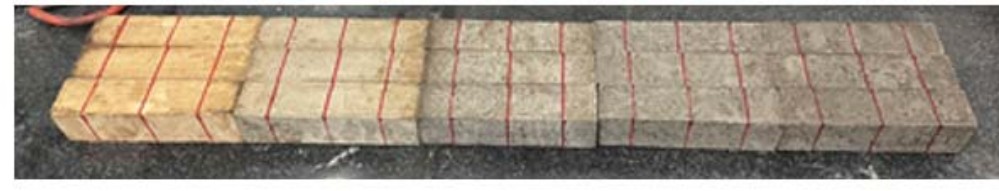

**Figure 3.** The pictures of specimens for different mix proportions.

### 2.2. Experimental Methods

The flowability and setting time of mortars were measured following GB/T 2419-2005 and BS-EN 196-3-2005. Compressive and flexural strength of the hardened mortars were measured using 40 × 40 × 160 mm specimens based on GB/T17671-1999. At least three specimens are tested for each group. The specimens were placed in the laboratory environment (20 ± 2 °C and 55% RH) until the time for the test. The flexural strength test was carried out by preloading at a rate of 0.2 mm/min to 20 N and then loading at 50 N/s until the specimen was fractured. The compressive strength test was carried out by load loading, first preloaded to 50 N and then formally loaded at a controlled rate of 2.4 KN/s.

Bonding properties were tested according to the Chinese standard JGJ/T 70-2009. The ordinary cement mortar (70.7 mm cubic) was cut into two pieces. A piece of ordinary cement mortar was placed vertically in the 70.7 mm cubic mold, then the BA-MPC mortar was poured in and mechanically vibrated. The specimens were tested for bonding properties at 28 days and the loading rate was 2.5 KN/s. The bonding strength ($\tau_u$) was calculated by the following formula:

$$\tau_u = \frac{P}{A} \tag{1}$$

where $P$ is the shearing load, MPa; the $A$ is the sectional area, mm$^2$.

The linear drying shrinkage was tested by length comparator according to the Chinese standard GB/T 50082-2009. Three mortar specimens with dimensions of 25 × 25 × 285 mm were prepared for each group. During the drying shrinkage testing period, the relative humidity and temperature were maintained at 55% and 20 °C.

An X-ray Diffractometer (XRD) was used to investigate the BA-MPC paste and the sample preparation was referred to [26]. The X-ray generator was operated at 40 mA and 40 kV. Scanning was performed from 5° to 80°. Thermogravimetry analysis was carried out using METTLER TGA/DSC 1. About 20–30 mg powder sample, which has been vacuum dried at 35 °C until achieved constant weight, was used for testing. The temperature range of thermogravimetry analysis was 30–800 °C and the heat rate was 10 °C/min under a nitrogen atmosphere. The pore structure of BA-MPC mortars measured by MIP and the sample preparation and test procedures were referred to [27]. The maximum pressure value of the high pressure was set as 61,000 psia. The microstructural characteristics of BA-MPC mortars were investigated by SEM. An acceleration voltage of 15 kV was used.

## 3. Results and Discussion

### 3.1. Flowability and Setting Time

Figure 4 shows the effect of FBA content on the flowability and setting time of MPC mortars. As FBA content increases, the flowability of MPC mortar first increases and then decreases. When the FBA content is low, the fill water is replaced due to the physical filling effect of FBA, which increases the free water to improve the flowability. When the water-to-binder ratio remains constant, further increased FBA content will lead to a reduction in the flowability of the mortar. This phenomenon is associated with that the poorer particle packing status due to excess FBA particles and the specific surface area of FBA is high than MPC resulting in the requirement for more water to wet the surface. The setting time decreases as FBA content increases. Because the MPC reaction raises the temperature of the paste, it promotes the reaction of $SiO_2$ in FBA with MgO. This will increase the ratio of phosphate ions to magnesium ions in the liquid phase, further promoting the reaction of MPC and resulting in a shorter setting time.

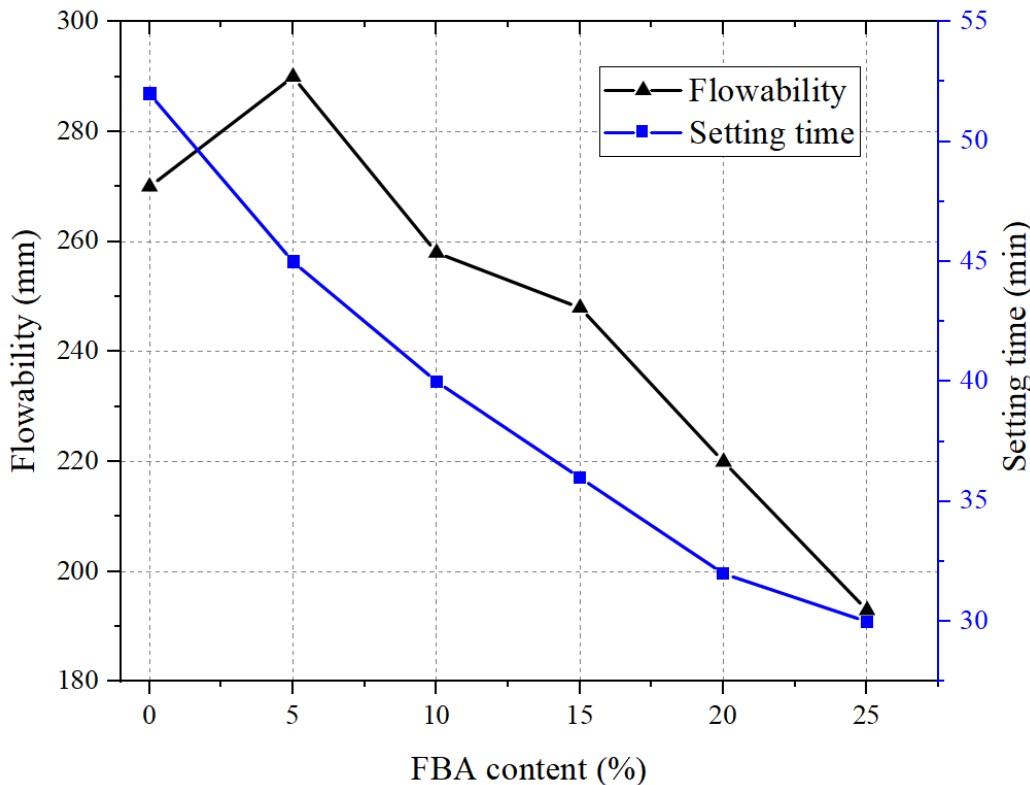

**Figure 4.** The effect of FBA content on the flowability and setting time of MPC mortars.

The effect of FBA and BBA on the flowability and setting time of MPC mortar (group 20 BA-80 MPC) is shown in Figure 5. Compared to the MPC mortar with FBA, the MPC mortar with BBA has higher flowability and longer setting time. The specific surface area of BBA is smaller than FBA, which results in higher flowability. Due to the low $SiO_2$ content and coarser particle size, BBA has much low reactivity resulting in lower hydration promotion for MPC and longer setting times of BA-MPC mortar.

### 3.2. Compressive and Flexural Strength

The effect of FBA content on the flexural strength of MPC mortars at different curing ages is shown in Figure 6. The flexural strength of MPC mortar decreases as FBA content increases. The degree of reduction in flexural strength is related to the curing age. When the curing age is 1 day, the reduction in flexural strength is in the range of 2–12%. The reduction in flexural strength gradually increases as the age of curing increases, ranging from 18%

to 40% when the curing age is 90 days. This indicates that the adverse effect of BA on the flexural strength of MPC mortar is aggravated with the increase of BA content and curing age. The effect of BA type on the flexural strength of MPC mortar (group 20 BA-80 MPC) is presented in Figure 7. The MPC mortar with 20% BBA has a lower flexural strength compared to the MPC mortar with 20% FBA, especially at longer ages. This is mainly due to the coarse particle size of BBA, which has a relatively low hydration activity.

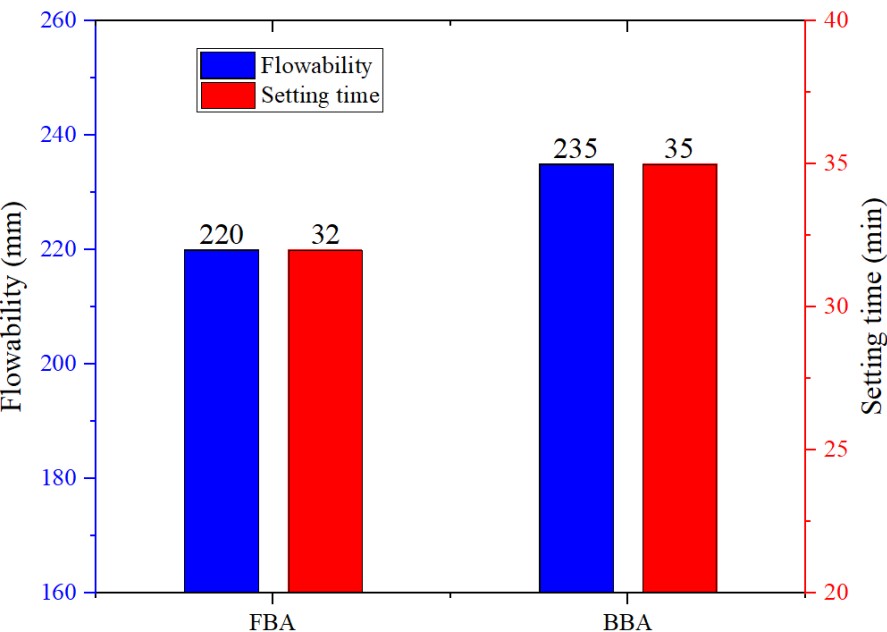

**Figure 5.** The effect of FBA and BBA on the flowability and setting time of MPC mortar (group 20 BA-80 MPC).

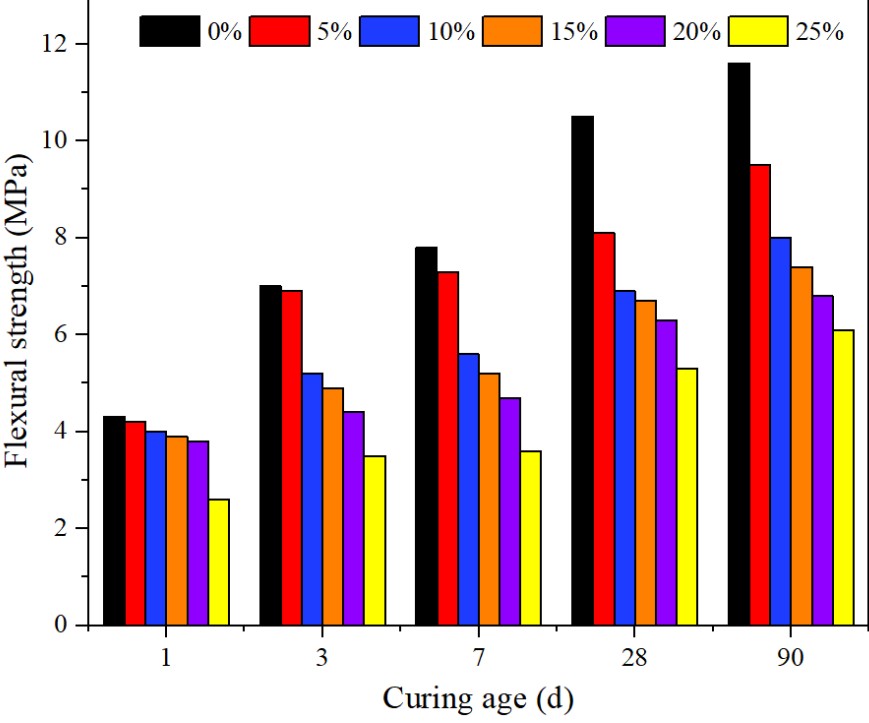

**Figure 6.** The effect of FBA content on the flexural strength of MPC mortars at different curing ages.

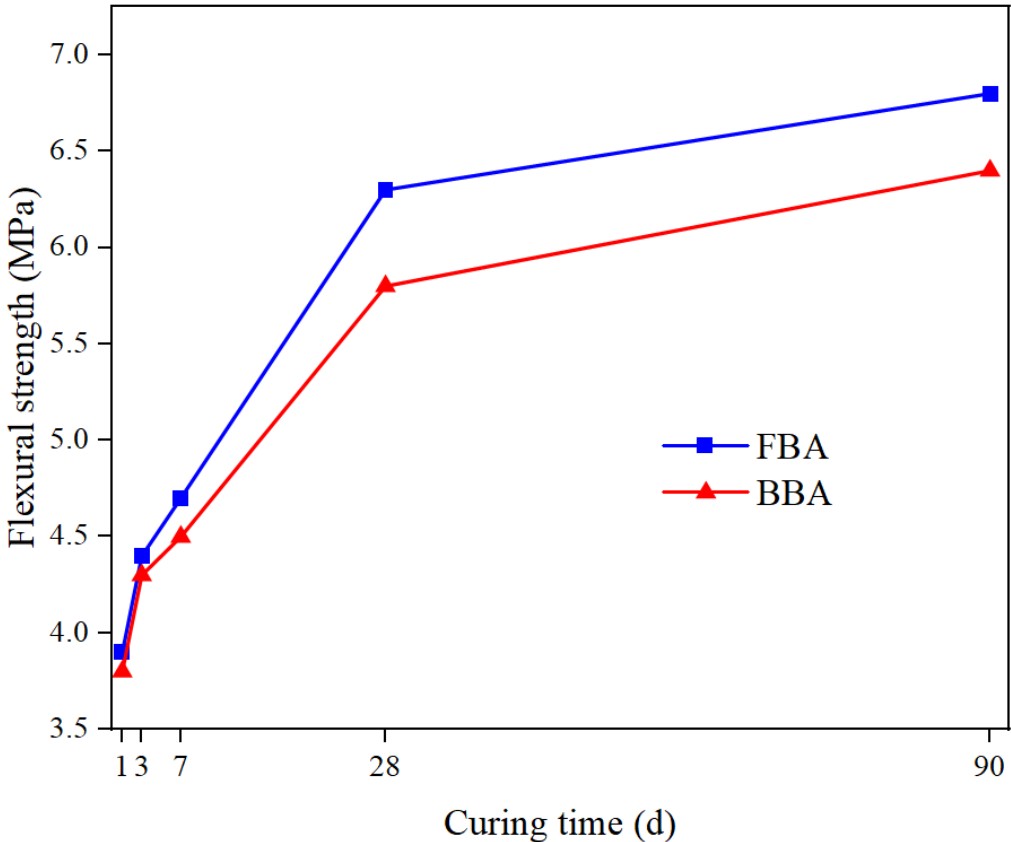

**Figure 7.** The effect of BA type on the flexural strength of MPC mortar (group 20 BA-80 MPC).

The effect of FBA content on the compressive strength of MPC mortars at different curing ages is shown in Figure 8. The effect of BA type on the compressive strength of MPC mortar (group 20 BA-80 MPC) is presented in Figure 9. By comparing Figures 7 and 8, the effect of FBA content on compressive strength is similar to that of flexural strength. When BA dosage is high, the incorporation of BA into MPC will result in low initial density, which contributes to a higher porosity. BA contains high amounts of $SiO_2$, which can react with MgO under certain conditions to form gel-like hydration products. At the beginning of the hydration reaction, the paste temperature rises significantly due to the violent reaction of the MPC, which promotes the reaction of BA with MgO. The shorter setting time of MPC mortar with BA is due to the reactive effect of BA. It is worth noting that the increase in temperature of the paste due to the reaction of MPC occurs over a relatively short period of time (Generally no more than 6 h) [28]. The reaction of BA at longer ages is not as high and makes a limited contribution to the number of hydration products. As the reactivity of the admixture is lower than cement, the dilution effect of the admixture will lead to a reduction in hydration products [29,30]. Therefore, the dilution effect of BA results in a reduction in MPC content in mortar leading to a lower volume of hydration products in a hardened paste. Those results in the long-term strength of the mortar showing a significant reduction with increasing amounts of BA.

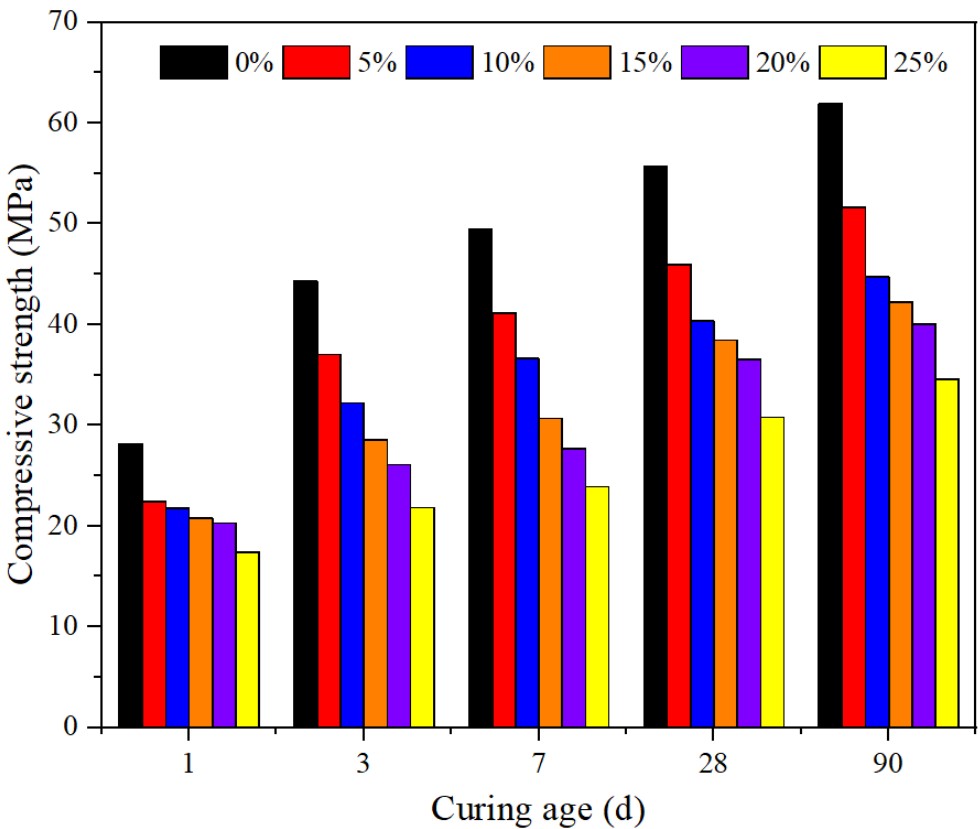

**Figure 8.** The effect of FBA content on the compressive strength of MPC mortars at different curing ages.

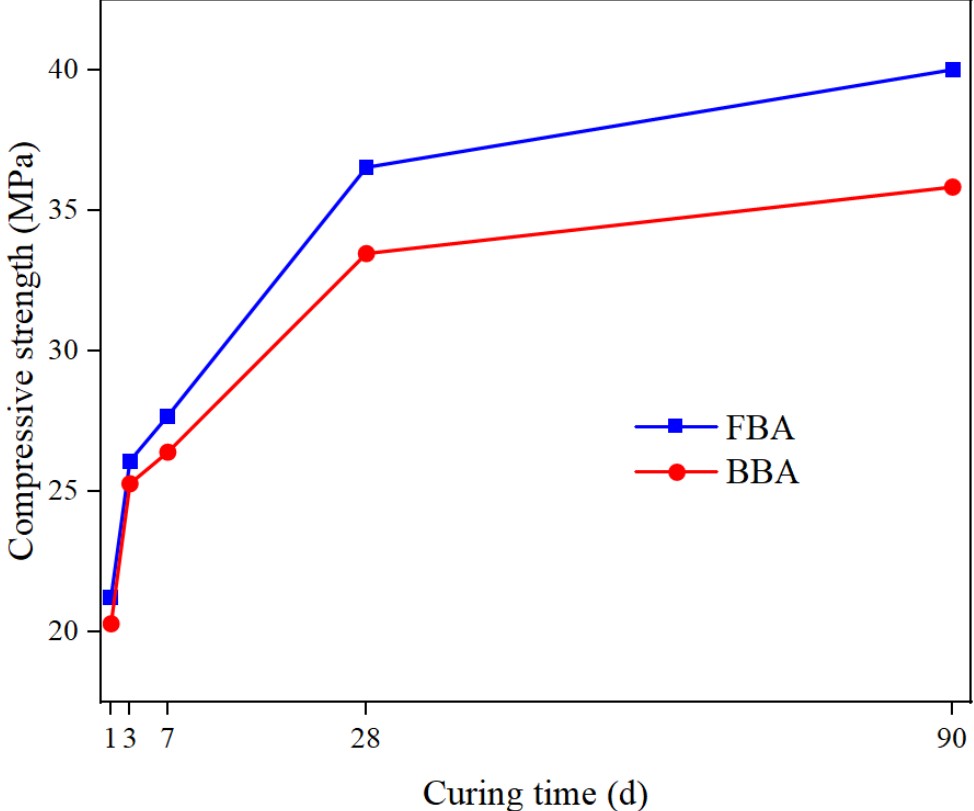

**Figure 9.** The effect of BA type on the compressive strength of MPC mortar (group 20 BA-80 MPC).

*3.3. Drying Shrinkage*

The results of drying shrinkage of MPC mortars with different FBA content are presented in Figure 10. As shown in Figure 10, the FBA content is an important factor in the drying shrinkage of MPC mortar. The drying shrinkage of MPC mortar specimens increases exponentially with the increase of FBA content. The increased rate of drying shrinkage at 1 d corresponded to 11%, 33%, 78%, 100%, and 156% when the FBA content increased from 5% to 25%, while that at 28 days corresponded to 4%, 9%, 20%, 30%, and 45%. This indicated that BA had a more significant effect on early-age drying shrinkage. An increase in the water-to-cement ratio will lead to an increase in drying shrinkage [31]. When cement is replaced by admixtures, the effective water-to-cement ratio of the paste increases due to the lower reactivity of the admixtures, which will lead to an increase in drying shrinkage [32]. As BA has a very low early reactivity at room temperature, it can be treated as an inert material at this stage. Therefore, the incorporation of BA in the MPC mortar causes a higher effective water-to-cement ratio and more free water in the paste due to the dilution effect of BA. Thus, more water was evaporated, especially at early ages, which will lead to higher drying shrinkage. Additionally, drying shrinkage is not only related to changes in relative humidity due to the evaporation of water, but also the pore structure of the material [33]. The incorporation of BA in the MPC mortar causes coarsening of the pores and an increase in the porosity, which will lead to an increase the drying shrinkage. The easier evaporation of water may be the main reason for the larger drying shrinkage. This indicates that the problem of large drying shrinkage deformation of MPC mortar with BA can be improved by changing the binder composition or curing regime, for example by adding admixtures to improve the reactivity of BA or by adding moisturizing measures.

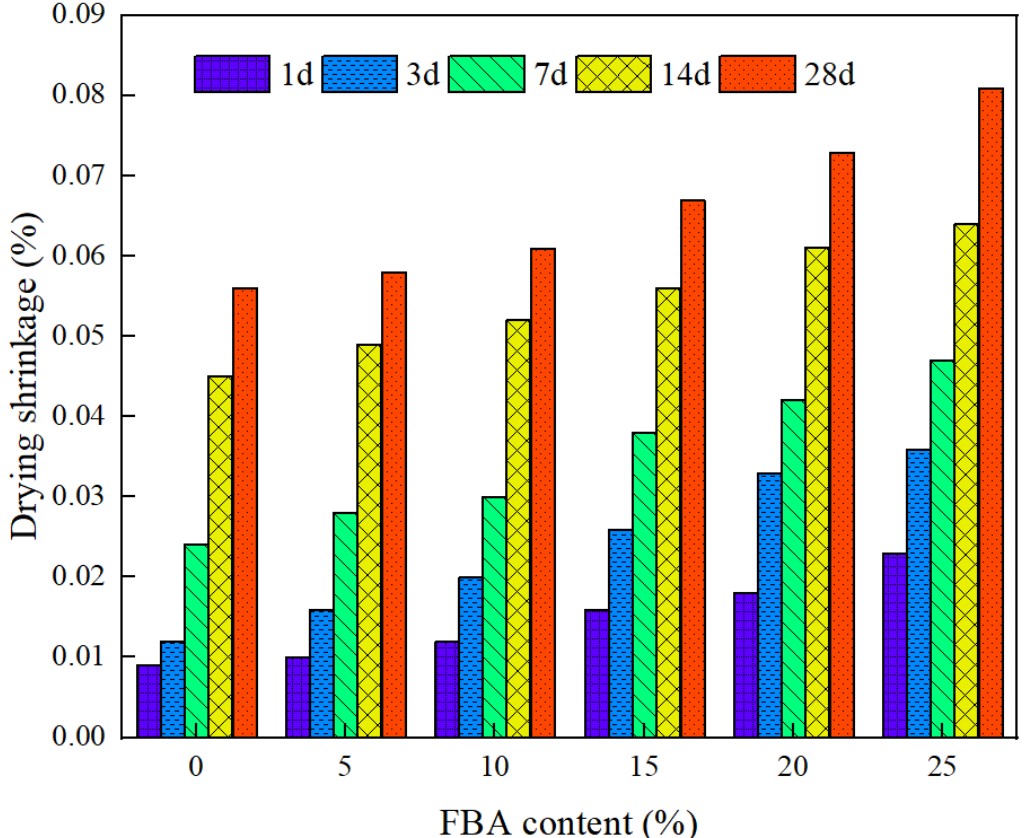

**Figure 10.** The results of drying shrinkage of MPC mortars with different FBA content.

Figure 11 illustrates the effect of BA type on the drying shrinkage of MPC mortar (group 20 BA-80 MPC). It is observed that the type of BA has a significant effect on the

drying shrinkage of MPC mortar. The incorporation of BBA into MPC mortars will cause greater drying shrinkage than FBA, especially at longer ages. The BBA has a larger particle size, which results in a lower initial packing density of the paste. The coarser particle size and lower silica content of BBA result in lower reactivity. The initial packing density of paste and reactivity of binder materials will influence the formation and development of the pore structure. The reactivity of binder materials will also influence moisture loss due to drying. Therefore, the MPC mortar with BBA has poorer porosity of hardened paste and higher drying moisture loss due to the physical and chemical properties of BBA, which leads to high drying shrinkage.

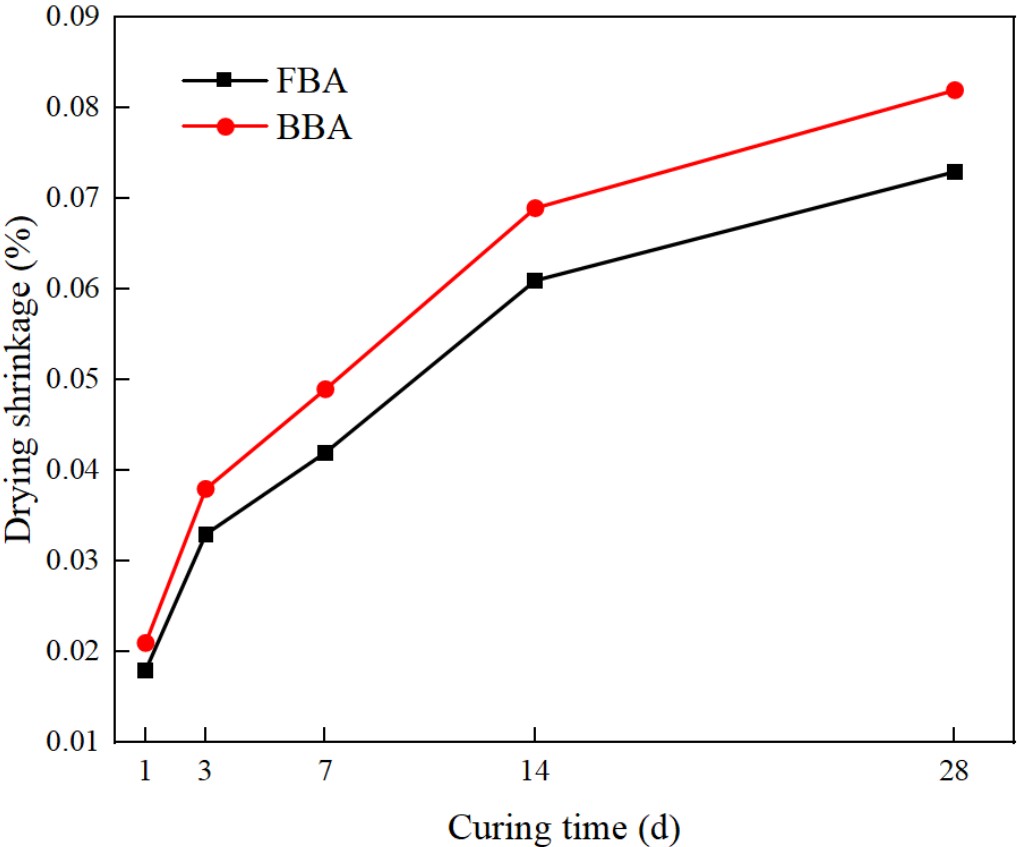

**Figure 11.** The effect of BA type on the drying shrinkage of MPC mortar (group 20 BA-80 MPC).

### 3.4. Bonding Strength

The bonding strength results of MPC mortars with different FBA content at 28 days are presented in Figure 12. The bonding strength of MPC mortar specimens showed a linear decrease with increasing FBA content. This coincides with the pattern of influence of FBA on compressive and flexural strength. Based on the results of Section 3.3, BA partially replacing MPC increases the drying shrinkage, which will have a negative impact on the bonding properties. In addition, the reduction in hydration products caused by the diluting effect of BA is also responsible for the reduction in bonding strength.

### 3.5. Microstructural Investigations

#### 3.5.1. XRD Analysis

Figure 13 shows the XRD patterns of BA-MPC paste samples (group 20 BA-80 MPC) at 28 days. The main crystalline substances in the MPC paste without BA are MgO and $MgKPO_4 \cdot 6H_2O$ (MKP). It can be found that the main characteristic peaks of the MPC samples with BA are generally consistent with the blank group. This means that the incorporation of BAs into MPC did not generate significant changes in the composition of the hydrated phases. The intensity of MgO means peaks are weaker when BA is

incorporated into MPC paste. This phenomenon is related to the dilution effect and reactivity effect of BA. The amount of unreacted MgO in MPC with BBA is higher than that of FBA. This s is related to the lower reactivity of BBA due to the larger particle size. The $MgKPO_4 \cdot 6H_2O$ is the main reaction product of MPC [34]. The amount of $MgKPO_4 \cdot 6H_2O$ produced is lowest in MPC with BBA and highest in MPC without BA. The BA, which has high $SiO_2$ content, will react with MgO to form a magnesium silicate gel [14]. The magnesium silicate gel is difficult to detect by XRD due to its amorphous nature. Therefore, the effect of the type of BA on the generation of hydration products also needs to be analyzed in combination with thermogravimetry analysis.

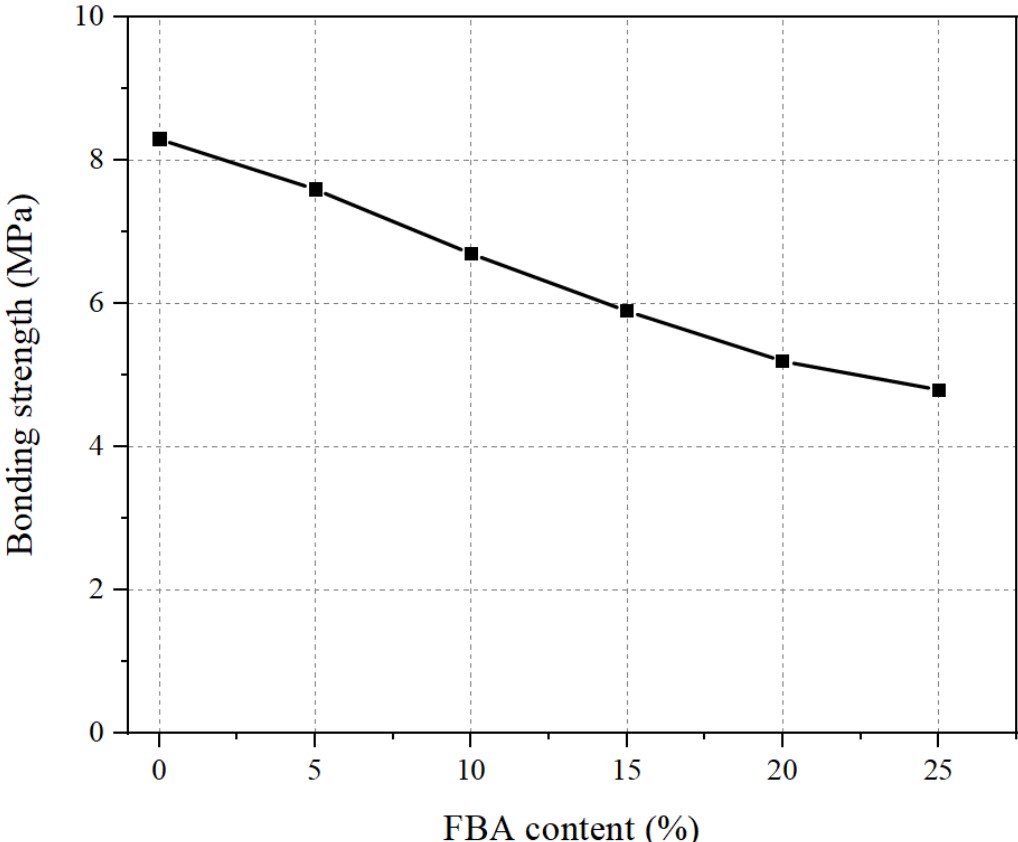

**Figure 12.** Bonding strength results of MPC mortars with different FBA content at 28 days.

### 3.5.2. Thermogravimetry Analysis

The TGA curves of MPC paste with or without 20% BA at 28 days are presented in Figure 14. The mass loss event at around 100 °C is mainly attributed to the evaporation of the crystalline water of $MgKPO_4 \cdot 6H_2O$ [35]. For the MPC paste with BA, the magnesium silicate gel produced by the reaction of BA with MgO also loses its free water at around 100 °C [14], resulting in an increased mass loss in this range. BBA group has an additional mass losses peak between 350–490 °C which is due to the loss of water from $Mg(OH)_2$ [Mechanisms of k-struvite formation in magnesium phosphate cement]. The mass losses near 100 °C for MPC without BA, MPC with 20% FBA, and MPC with 20% BBA are 23.14%, 21.38%, and 17.13%, respectively. This indicates that although BA has reactivity, the incorporation of BA into MPC still leads to a reduction in hydration products. In addition, the FBA group has a higher amount of hydration products than the BBA group due to its higher hydration activity. Those can also explain the phenomenon that the strength of the designed BA-MPC mortars follows the same order: 20 BBA-80 MPC < 20 FBA-80 MPC < 100 MPC.

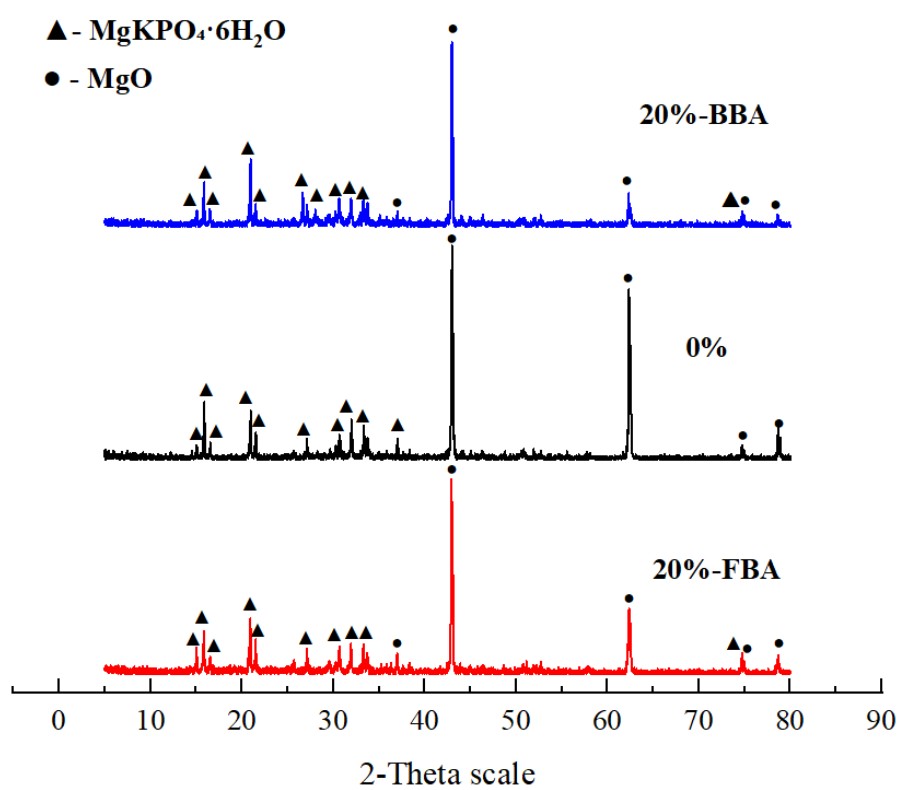

**Figure 13.** The XRD patterns of BA-MPC paste samples (group 20 BA-80 MPC) at 28 days.

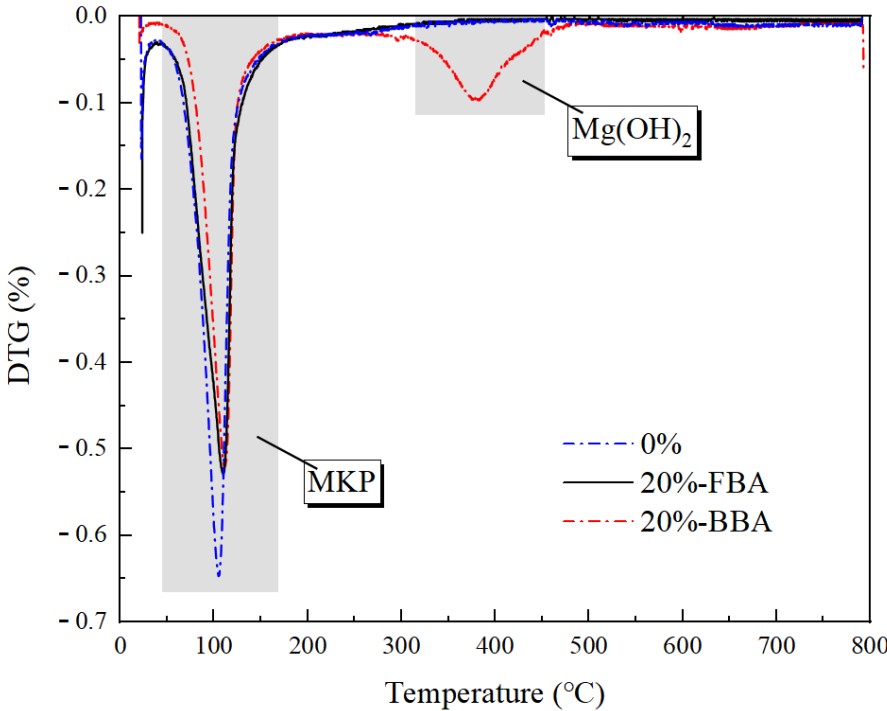

**Figure 14.** The TGA curves of MPC paste with or without BA at 28 days.

### 3.5.3. Pore Structure Analysis

The results of the pore structure of MPC mortars with or without 20% BA at 28 days by MIP are shown in Figure 15. The test results clearly indicated an increase in porosity and average pore diameter when incorporating BA into the MPC. When BA is mixed in MPC at high levels, it results in a poorer particle packing state. BA has a lower reactivity resulting

in fewer hydration products in the hardened paste. These together lead to a coarsening of the pore structure of MPC mortars. In comparison to BBA, the incorporation of FBA into MPC has lower porosity and a smaller average pore diameter. This clearly indicates that BA with smaller particle sizes and higher silica content will result in MPCs with more hydration products and better pore structure. Therefore, the selection of BA with a high silica content and grinding them to a smaller particle size will improve the suitability of BA for use in MPC.

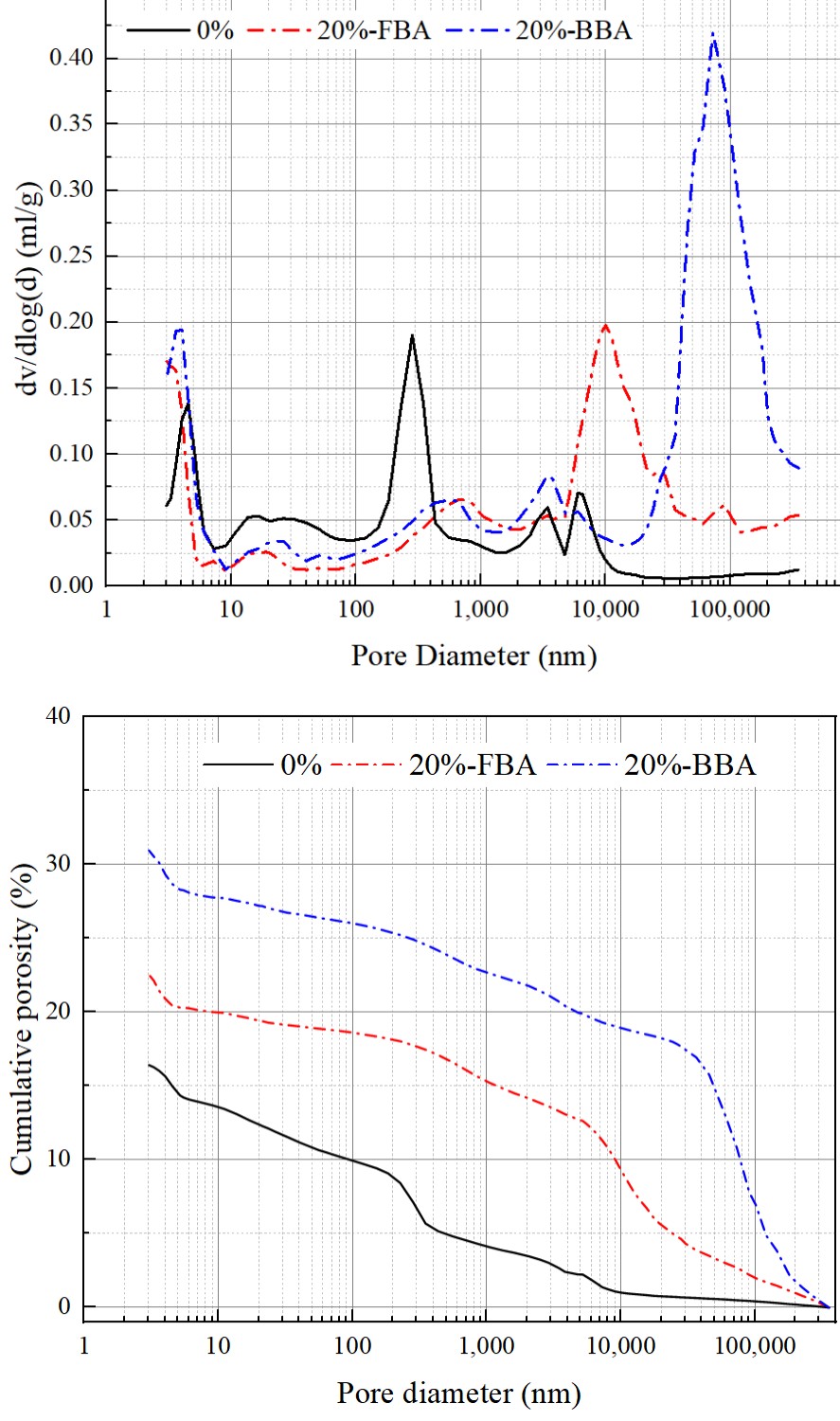

**Figure 15.** The results of pore structure of MPC mortars with or without BA at 28 days by MIP.

### 3.5.4. SEM Analysis

Figures 16 and 17 show SEM micrographs of MPC without BA and MPC with 20% FBA specimen at 28 days. As shown in Figures 16 and 17, the incorporation of BA into MPC has a significant effect on the microstructure morphology. When BA is not added, the main hydration product is MKP, and a layer sheet product is distributed on the surface. The hollow columnar-like hydration product may be formed by the reaction of BA with MgO in the paste.

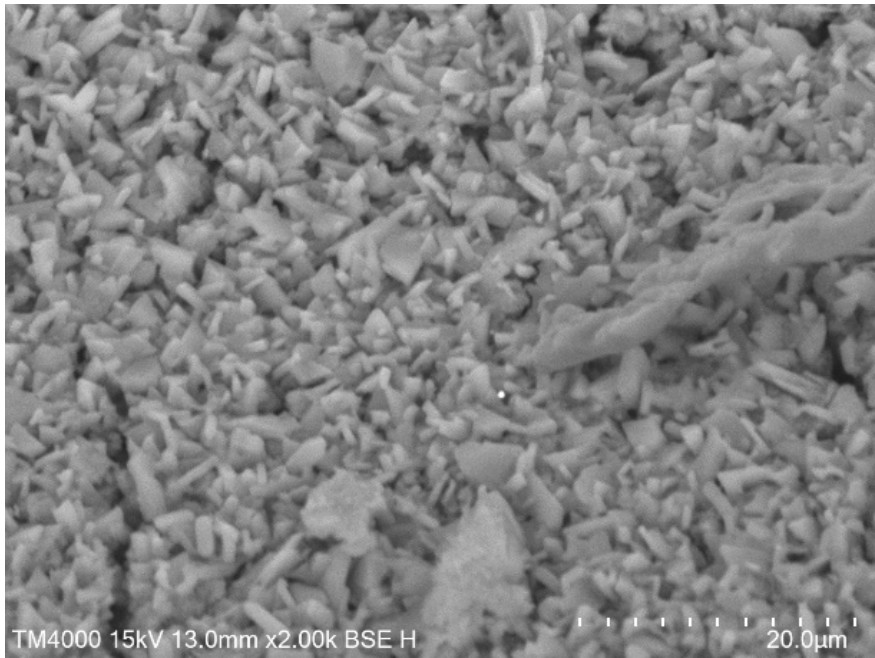

**Figure 16.** SEM micrographs of MPC without BA at 28 days.

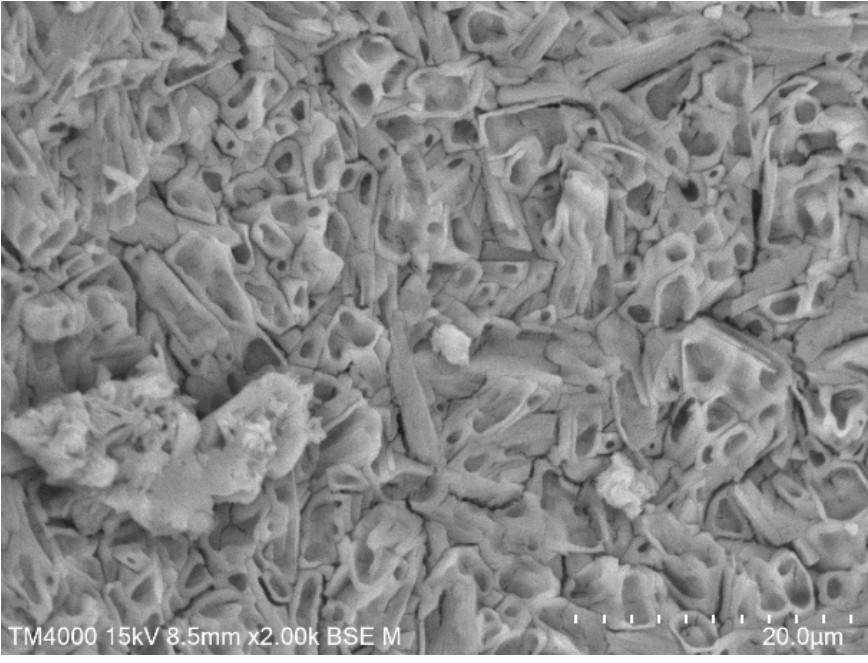

**Figure 17.** SEM micrographs of MPC with 20% BA at 28 days.

## 4. Conclusions

1.  The flowability of MPC mortars first increased and then decreased as BA content increased. The setting time of MPC mortars decreased as BA content increased due to the reaction of $SiO_2$ in BA with MgO.

2.  The compressive and flexural strength of MPC mortars decreased with increasing amounts of BA. The effect of BA on the strength was not significant when the BA dosage is low and the curing age is short. The drying shrinkage of MPC mortar specimens increased exponentially with the increase of BA content. The incorporation of BA reduced the bonding strength of the MPC mortar, which is associated with increased drying shrinkage.

3.  XRD results showed that no new crystalline hydration products were formed when BA was incorporated into MPC and the main crystalline hydration product was $MgKPO_4 \cdot 6H_2O$. In combination with thermogravimetric analysis, it is confirmed that BA incorporation produces a certain amount of magnesium silicate gel. Although BA has reactivity, the incorporation of BA into MPC still led to a reduction in hydration products.

4.  The MIP test results showed an increase in porosity and average pore diameter when incorporating high-volume BA into the MPC, which was associated with the poor particle packing state and low hydration product generation. The incorporation of BA into MPC had a significant effect on the microstructure morphology and the hollow columnar-like hydration product may be formed by the reaction of BA with MgO in the paste.

**Author Contributions:** Conceptualization, S.Z. and P.W.; methodology, Y.S.; software, Y.S.; validation, P.W.; investigation, S.Z.; resources, S.Z.; data curation, H.Z.; writing-original draft preparation, S.Z. and Y.S.; writing-review and editing, P.W.; visualization, Y.H.; supervision, W.J.; project administration, P.W. All authors have read and agreed to the published version of the manuscript.

**Funding:** This research was funded by the Natural Science Foundation of Tianjin, China grant number 19YFZCSN01140.

**Institutional Review Board Statement:** Not applicable.

**Informed Consent Statement:** Not applicable.

**Data Availability Statement:** Not applicable.

**Acknowledgments:** The test result was finished in Tianjin Key Laboratory of civil structure protection and reinforcement, Tianjin Chengjian University.

**Conflicts of Interest:** The authors declare no conflict of interest.

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
