# Peer review of "Effect of Biomass Ash on the Properties and Microstructure of Magnesium Phosphate Cement-Based Materials"

_buildings, doi:10.3390/buildings13010030_

Round 1
Reviewer 1 Report
In this paper, the different properties of magnesium phosphate cement with biomass ash were studied. The research content has some novelty. There are many research methods. It is recommended to consider the following opinions and suggestions.
In Figure 2, the maximum particle size reaches 100mm? How to stir in this case? How to test particle size? Establish and list the specific test process.
In Figure 11, how is the bond strength tested? Establish and list specific processes.
In Figure 14, it is recommended to supplement the cumulative mercury inflow curve.
It is recommended to add pictures corresponding to different mix proportions.
It is recommended to consider the following references:
Effect of biomass power plant ash on fresh properties of cemented coal gangue backfill.
https://www.sciencedirect.com/science/article/abs/pii/S0950061822015264
Effect of biomass Ash, foundry sand and recycled concrete aggregate over the strength aspects of the concrete
https://www.sciencedirect.com/science/article/pii/S2214785321063215
Synergistic effects of three-dimensional graphene and silica fume on mechanical and chloride diffusion properties of hardened cement paste
https://www.sciencedirect.com/science/article/abs/pii/S0950061821034899
Author Response
Dear reviewer:
Thank you for providing positive and constructive comments and suggestions for our manuscript. We hope this revised manuscript would finally meet your requirements. Thank you once again and best regards.
Comment 1: In Figure 2, the maximum particle size reaches 100mm? How to stir in this case? How to test particle size? Establish and list the specific test process.
Response: The particle size should be in µm and has been corrected in the Figure 2. The particle size distribution of BAs was measured by laser diffraction method. The test process has been added to the manuscripts.
Comment 2: In Figure 11, how is the bond strength tested? Establish and list specific processes.
Response: The specific processes of bond strength test have been added to section 2.2 as suggested.
Comment 3: In Figure 14, it is recommended to supplement the cumulative mercury inflow curve.
Response: The Cumulative porosity curve have been added to Fig.14 as suggested.
Comment 4: It is recommended to add pictures corresponding to different mix proportions.
Response: The pictures corresponding to different mix proportions have been added to section 2.1 (Fig.3) as suggested.
Comment 5: It is recommended to consider the following references:
Effect of biomass power plant ash on fresh properties of cemented coal gangue backfill.
https://www.sciencedirect.com/science/article/abs/pii/S0950061822015264
Effect of biomass Ash, foundry sand and recycled concrete aggregate over the strength aspects of the concrete
https://www.sciencedirect.com/science/article/pii/S2214785321063215
Synergistic effects of three-dimensional graphene and silica fume on mechanical and chloride diffusion properties of hardened cement paste
https://www.sciencedirect.com/science/article/abs/pii/S0950061821034899
Response: Relevant references have been added to the article as suggested.
Reviewer 2 Report
-We live now in a climate emergency so its most strange that the authors have not start the paper by mentioning exactly that. It seems that they are not aware about the words of a Professor of Physics at the University of Oxford authored a paper where one can read the following:
“Let’s get this on the table right away, without mincing words. With regard to the climate crisis, yes, it’s time to panic”
Pierrehumbert, R., 2019. There is no Plan B for dealing with the climate crisis. Bulletin of the Atomic Scientists, pp.1-7.
So please start the introduction by draw a connection between environmental degradation and resource efficiency.
- “As the world's fourth largest energy source, biomass energy is both green, low-carbon and renewable [1]”
Comment: Reference 1 is not a study that assess the environmental footprint of biomass. Check the highly cited paper below that shows biomass as a rather high environmental footprint
Hadian, S., & Madani, K. (2015). A system of systems approach to energy sustainability assessment: Are all renewables really green?. Ecological Indicators, 52, 194-206.
- The first phrase of the abstract must be a sumary of the introduction
- “there is only a little research on the application of BA in civil engineering”
Comment: What search words did the authors of this paper used to reach that conclusion ?
Author Response
Dear reviewer:
Thank you for providing positive and constructive comments and suggestions for our manuscript. We hope this revised manuscript would finally meet your requirements. Thank you once again and best regards.
-We live now in a climate emergency so its most strange that the authors have not start the paper by mentioning exactly that. It seems that they are not aware about the words of a Professor of Physics at the University of Oxford authored a paper where one can read the following:
“Let’s get this on the table right away, without mincing words. With regard to the climate crisis, yes, it’s time to panic”
Pierrehumbert, R., 2019. There is no Plan B for dealing with the climate crisis. Bulletin of the Atomic Scientists, pp.1-7.
So please start the introduction by draw a connection between environmental degradation and resource efficiency.
Response:
We are grateful to the reviewers for their comments, which have provided us with a deeper understanding of the climate emergency. As this article is primarily a study of the resource use of solid waste, with limited relevance to the climate emergency, only a brief mention of this section will be included.
- “As the world's fourth largest energy source, biomass energy is both green, low-carbon and renewable [1]”
Comment: Reference 1 is not a study that assess the environmental footprint of biomass. Check the highly cited paper below that shows biomass as a rather high environmental footprint
Hadian, S., & Madani, K. (2015). A system of systems approach to energy sustainability assessment: Are all renewables really green?. Ecological Indicators, 52, 194-206.
Response:
Thanks to the reviewers' comments, this gives us a better understanding of biomass energy. After reading this paper, we also think that this paper is more appropriate. Therefore, we have partially revised the introduction and replaced it in the references.
- The first phrase of the abstract must be a sumary of the introduction
Response:
Thanks to the reviewers' comments, the abstract has been revised, which will better highlight the topic of the study.
- “there is only a little research on the application of BA in civil engineering”
Comment: What search words did the authors of this paper used to reach that conclusion ?
Response:
Biomass ash (BA) and magnesium phosphate cements (MPC) were used as the main keywords respectively. The aim achieved through the literature summary is: on the one hand, to understand the application of biomass ash in the field of civil engineering; on the other hand, to understand the current state of research on magnesium phosphate cements, with particular attention to the study of the application of various mineral admixtures in magnesium phosphate cement systems. According to the summary of some previous studies, the application of BA is only limited to the study of the influence of MCP on the strength, without in-depth understanding of its influence principle.
Round 2
Reviewer 1 Report
The author has certified and revised the paper, which is acceptable.